# Decoding the diagnostic and therapeutic potential of microbiota using pan-body pan-disease microbiomics

The human microbiome emerges as a promising reservoir for diagnostic markers and therapeutics. Since host-associated microbiomes at various body sites differ and diseases do not occur in isolation, a comprehensive analysis strategy highlighting the full potential of microbiomes should include diverse specimen types and various diseases. To ensure robust data quality and comparability across specimen types and diseases, we employ standardized protocols to generate sequencing data from 1931 prospectively collected specimens, including from saliva, plaque, skin, throat, eye, and stool, with an average sequencing depth of 5.3 gigabases. Collected from 515 patients, these samples yield an average of 3.7 metagenomes per patient. Our results suggest significant microbial variations across diseases and specimen types, including unexpected anatomical sites. We identify 583 unexplored species-level genome bins (SGBs) of which 189 are significantly disease-associated. Of note, the existence of microbial resistance genes in one specimen was indicative of the same resistance genes in other specimens of the same patient. Annotated and previously undescribed SGBs collectively harbor 28,315 potential biosynthetic gene clusters (BGCs), with 1050 significant correlations to diseases. Our combinatorial approach identifies distinct SGBs and BGCs, emphasizing the value of pan-body pan-disease microbiomics as a source for diagnostic and therapeutic strategies.

Recent advancements in the field of microbiota research have spotlighted the complex interplay between non-communicable diseases and the human microbiome, offering novel avenues for understanding disease pathogenesis, identifying biomarkers, and developing new therapies[1–8]. Respective studies demonstrate site-specific changes in microbiota linked to disease development and progression. Large-scale metagenomic initiatives, such as the Human Microbiome Project, International Human Microbiome Consortium[9], the American Gut Project, and other hallmark studies[10–16], have compiled thousands of microbiota samples leading to results arguing all in the very same direction. Data on the impact of an organ-confined disease process in the microbiota of remote body sites remain, however, sparse. Understanding the dynamic relationship between health and disease,

especially in the context of frequent co-morbidities, necessitates a holistic exploration of the human microbiome. One example is the discovery of the gut-brain axis, a refined communication between the intestinal microbiota, intestinal host cells, and the central nervous system via the vagal nerve[17]. Overall, the microbiome can be modified at sites remote from the primary disease process[18,19]. Thus, understanding the composition of metagenomes in health and disease in a systemic manner might be key to improving patient care.

Especially considering the growing health challenges spurred by demographic shifts and social influences, the imperative for enhanced treatment strategies becomes increasingly evident. For example, chronic inflammatory diseases encompass a diverse array of conditions such as periodontitis, chronic obstructive pulmonary disease

✉e-mail: andreas.keller@ccb.uni-saarland.de

(COPD), cystic fibrosis, cardiovascular diseases, heart failure, and ulcerative colitis, among others[20–25]. Respective ailments collectively pose a significant burden on individuals and healthcare systems alike. This calls for an in-depth exploration of underlying molecular mechanisms[26,27]. These and other non-communicable diseases are associated with chronic local and systemic inflammation and often occur as multimorbidities. Processes driving the development of multimorbidity are largely unknown but include a state of systemic hyperinflammation, metabolic changes, and senescence. While the underlying mechanisms leading to these (co)morbidities are not yet fully explored, emerging evidence suggests that chronic inflammatory diseases are intrinsically linked to perturbations in the composition and function of the human microbiota[28]. Imbalances in microbial communities residing at various body sites, including the oral cavity, respiratory tract, gastrointestinal tract, and skin, have been implicated in the initiation and perpetuation of inflammatory cascades[29–32]. This has led to the paradigm-shifting realization that the microbiota, once considered a bystander, plays a pivotal role in disease progression and resolution.

The pharmaceutical landscape has long drawn inspiration from nature, with a significant - but reducing - proportion (35% percent) of drugs on the market derived from natural products (NPs) and their producers. In 2021, 50 drugs were approved by the FDA[33]. Of those, 14 approvals represent biologics and 36 small molecules. Only four of the drugs are based on NPs which do not cover some of the greatest needs in human medicine, indicating the need to improve strategies to identify new NPs. These NPs are typically encoded by so-called biosynthetic gene Clusters (BGCs). Considering specific microbial interactions and their underlying chemical processes promises to identify new NPs that remain hidden in classical culture-based studies. Although most isolates are classified as commensals, the heightened prevalence of specific strains, notably methicillin-resistant *S. aureus* (MRSA), constitutes a substantial risk factor for severe and frequently fatal infections and finally led to epifadin as a new antimicrobial compound class. In a recent investigation, a novel dimension of microbial interactions was unveiled through the examination of the interplay between *Staphylococcus epidermidis* and *S. aureus* isolated from the human nasal region[34].

Highly accurate, fast, and inexpensive next-generation sequencing paired with computational tools like antiSMASH[35,36] are now used at scale in genome mining approaches to identify novel BGCs from microbial communities. As a consequence, important resources such as the BiG-FAM[37] database are increasing in size and complexity, now hosting 1.2 million BGCs, and being potentially the dominating source for BGCs. However, the annotation of microbiota-derived BGCs originating from humans, especially in the context of BGCs present at different body sites, is largely absent, presenting a gap in evidence-based prioritization strategies aimed at identifying BGCs with the highest therapeutic potential. As an initial effort to address this gap, we developed ABC-HuMi[38], a database featuring BGCs identified across various human body sites. Nevertheless, to provide a systematic disease context for diverse specimen types and diseases, data e.g. from meta-analyzes might not be sufficient. Here, metagenomes from deeply phenotyped cohorts must be analyzed in a standardized manner.

The aim of this study is to fill current gaps by comprehensively characterizing the pan-body alterations of the microbiome in single-organ disease and multimorbidity. The inclusion of different diseases is crucial in several aspects. The analysis of multiple disorders allows understanding the specificity of different abundances of species and BGCs for single diseases[39]. At the same time, it tremendously improves the definition of a "healthy" versus "normal" microbiome. Admittedly, the combination of multiple diseases affecting multiple organs and collecting multiple specimen types per patient bears significant

challenges in including patients, measuring the sample with the least possible bias, and of course computational complexity and interpretation. Yet, our study provides steps forward in the ability to identify new diagnostic and therapeutic strategies as the basis for AI tools with increasing importance in NP development[40].

## Results

### Standardized workflow considering multi-morbid patients

Our study results are derived from clinical, experimental, and computational components (Fig. 1a). Between 2021 and 2023 we collected a total of 3483 samples from 657 individuals spanning a broad spectrum of diseases (chronic inflammatory diseases of the lung, heart, eyes, intestine, skin, and oral cavity). Nine different departments at the Saarland University Hospital (dentistry, dermatology, cardiology, gastroenterology, pulmonology, ophthalmology, pediatry, periodontology, and sports medicine) recruited patients. The standardized sample collection and medical assessment across disciplines (e.g., each patient, independently of the enrolling clinics, medical assessment, and oral examination) was a key criterion within the clinical workflow. The enrolling sites transferred the specimens to the Department of Microbiology for biobanking and metagenomic sequencing and stored medical data in a database after manual curation. A respective approach specifically requires standardized protocols for metagenomic sequencing that we previously developed[41]. The computational analysis team obtained both, the raw sequencing data as well as the clinical information after blinded measurement of the metagenomes. We analyzed the microbial composition of multiple body sites, including the oral cavity (saliva, interdental plaque), the skin, the throat swabs, the gastro-intestinal tract (stool), and the eye (conjunctiva swabs), largely matching the affected organs (Fig. 1b). The standardized sequencing results combined with broad and well-curated medical annotations are the basis for the computational analyzes, linking known and not yet annotated bacterial species and their repertoire of BGCs to human pathologies, finally representing a unique resource for new NPs.

### Metagenomic sequencing and clinical annotation yield 1931 high-quality samples

While the perfect world scenario encompasses a complete dataset where every patient's sample is collected and each sample results in a high-quality (HQ) metagenome, in contrast, the real-world situation presents a more complex scenario. As not all patients consented to collecting all specimen types, and not all collected specimens yielded HQ metagenomes, we applied stringent quality control to avoid bias in the analysis of profiles. From the 3483 collected samples, we obtained 1931 HQ metagenomes after quality assessment. These stem from 515 individuals, marking an average of 3.7 microbiomes for each proband (Supplementary Data 1). We excluded samples for various reasons, the most important being an insufficient amount or quality of DNA (c.f. Methods, Fig. 1c). The sample removal process was unevenly distributed, with significantly lower losses of stool, saliva, plaque, and throat specimens, and conversely significantly increased loss of skin and eye specimens. For each metagenome passing the quality control assessment, regardless of the specimen type, we obtained an average of 5.3 gigabases (standard deviation +/− 2.3 gigabases) of metagenomic information after removing ambient human DNA (Fig. 1d). Together, the sequencing efforts generated 10.2 terabases of non-human sequencing information. We used this curated and annotated dataset of 1931 metagenomes and clinical data through the upcoming analyzes. This dataset excels in that over 90% of all samples belong to a patient with at least three different metagenomes available (Supplementary Fig. 1a). As a first analysis, we considered the observed comorbidity pattern of the enrolled individuals (Fig. 1e). The majority of 437 individuals were patients, i.e. probands diagnosed with at least

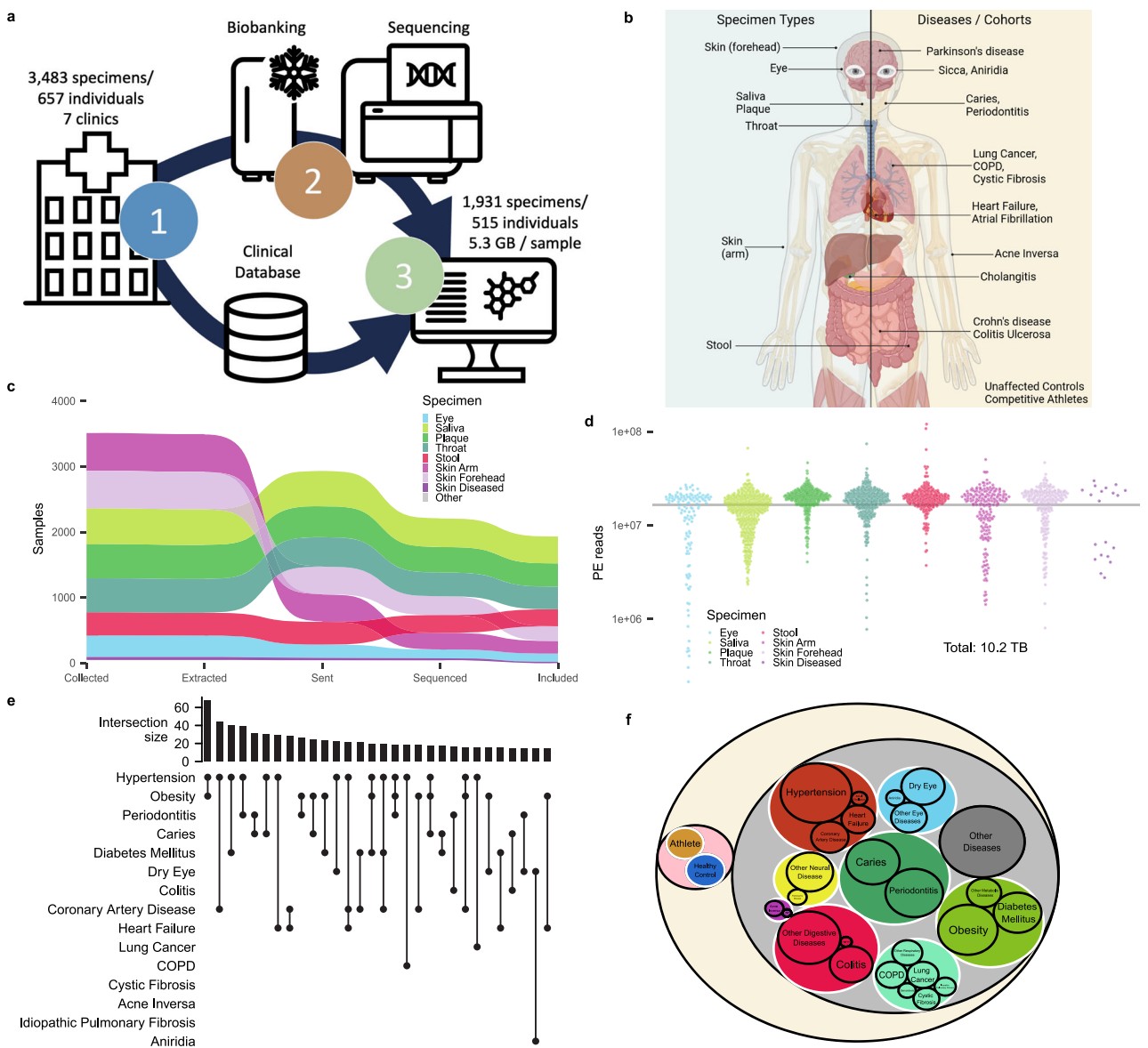

**Fig. 1 | Study set up, metagenomics data and clinical information. a** Schematic Workflow describing the sample (upper arrow) and data flow (lower arrow) between clinicians, microbiology, and data science. The clinical data were kept separated from the measurement of microbiomes and only combined after measurement in the computational analysis. **b** Clinical sampling was focused on seven biospecimens (left blue part). We included patients from a wide range of clinical diseases that allows us analyzing the diagnostic potential of different specimen types across diseases. Created with BioRender.com released under a Creative Commons Attribution-NonCommercial-NoDerivs 4.0 International license. **c** Sankey plot for the number of samples included in the study at different intervals of the data generation process in relation to our quality control strategy. Specimen types are ordered vertically at each step in the pipeline by frequency of the respective specimen. **d** Number of reads for each sample colored by specimen. The horizontal line represents the 5 gigabase threshold at a paired-end read length of 150 bp. **e** Pruned upset plot displaying the most frequent co-occurrence of diseases within the dataset. The combinations are ordered with decreasing frequency, marking the combination of Hypertension and obesity as most common comorbidity in our study. **f** Ontology used throughout the study grouping diseases by biological systems and separating healthy control from diseased patients. Areas are proportional to the number of patients falling into each category. Patients may be represented multiple times if multiple diseases are diagnosed.

one disease. We included further 46 participants as controls, encompassing individuals without known disease affection and an additional 36 competitive athletes. The five most frequent disease entities in the cohort were hypertension (32.4%), obesity (21.7%), periodontitis (18.8%), caries (16.7%), and diabetes (14.2%). We assessed comorbidities, with many patients suffering from two (23.7%) or three (17.4%) concurrent diseases, respectively. The most frequently observed association was diabetes and cardiac failure. Of note, covering a broad spectrum of diseases intrinsically and intentionally leads to a broad age spectrum (Supplementary Fig. 1b). Similarly, we observed an

uneven gender distribution leaning towards more males than females, with a ratio of 58.1% to 41.9%. Together, these factors led to a slight yet statistically significant gender disparity persisting across different age groups ($\chi 2$ test $p$-value = 0.02). To reach sufficient statistical power we propose a three-tiered ontology framework for organizing our cohort analysis (Fig. 1f). By categorizing samples into hierarchical tiers, we ensured to enhance power while acknowledging the potential for increased heterogeneity and confounding factors within the data (Supplementary Data 2). We defined the associated cohort of a specimen as the second-level ontology category encompassing all diseases

logically linked to the disease itself; for instance, all oral diseases are associated with the plaque specimen. This second ontology level gave the best overall balance between specificity and still having sufficient samples in the respective cohorts. We thus used this level throughout the manuscript and explicitly mention when another ontology level is the basis of a result.

## Compositional analysis identifies a complex pattern of microbiome-disease associations

To get insights into the general distribution of metagenomes we computed an embedding based on the MinHash distances (Fig. 2a). This embedding highlighted the separation of metagenomes according to the specimen types, with samples from the oral cavity clustering

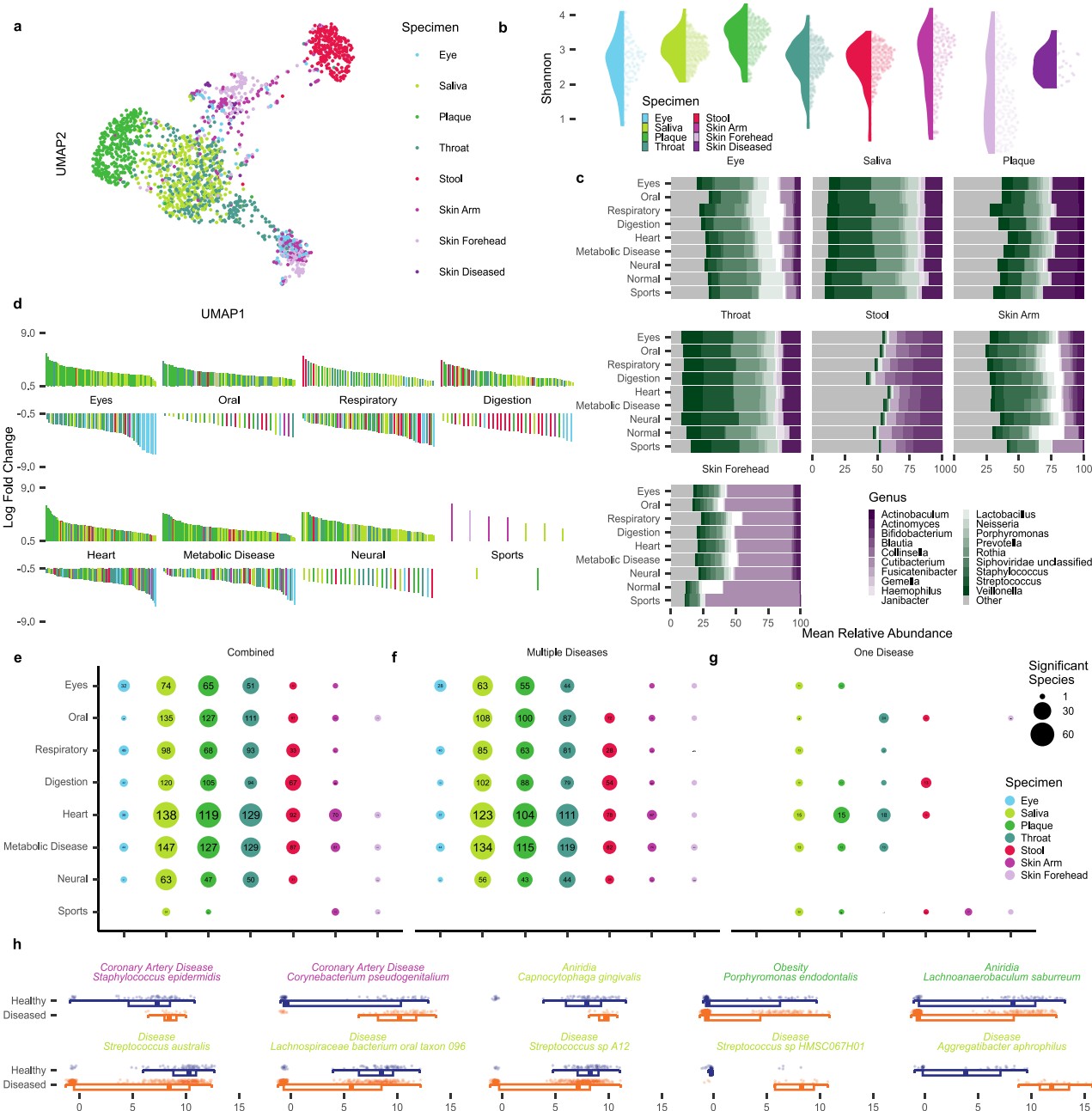

**Fig. 2 | Compositional analysis, and link of microbiota to diseases. a** Two-dimensional Uniform Manifold Approximation and Projection (UMAP) embedding of pairwise computed mash distances, colored by biospecimen of the sample. **b** Alpha-diversity of all samples, colored by specimen. As a measure of species richness, we selected the Shannon diversity. **c** Relative genus abundance for each cohort of the second ontology level, divided by biospecimen. Only labels for the 20 most abundant genera are displayed. **d** Sorted log-fold changes of differentially abundant species matching the visualized results of the next panel. Each panel is split vertically separating positive and negative log-fold changes. **e**–**g** Number of differentially abundant species after *p*-value adjustment of ANCOMBC results revealed during analysis across all cohorts and specimen combinations (q-val

<0.05). Numbers in the circles represent the number of specimens included in the respective analysis. **h** Center-log ratio (CLR) normalized abundance counts of selected species-cohort-specimen combinations. The visualized diseased cohort is indicated by the text above each panel, whereas the selected biospecimen is indicated by the color of the writing. The first row of panels displays potential pathogen candidates with the highest statistical significance and a pathogen score of one. The second row of panels displays saliva samples of commensal bacteria candidates with a commensal score larger than eighteen (min(n) = 50). Boxplot follows Tukey's style indicating the median as well as the second and third quantiles within boxes. Whiskers extend up to 1.5 times the interquartile range in the presence of outliers.

together and being most different from the stool samples. Samples with lower bacterial and DNA abundance from the skin and the eye clustered between these two groups, however splitting up in two separate clusters. Those two clusters have no common confounding factors such as sequencing batches, age, gender, and others, with the only difference being the duplication rate of reads (Two-tailed unpaired Wilcoxon $p$-value $< 10^{-41}$, Supplementary Fig. 2). To account for this effect in downstream analyzes, we adjusted our models for the duplication rate within samples. Taking the Shannon entropy as a measure for the diversity of the sequenced microbiota, we recognized a broader spread of the specimen types with lower DNA yield (Fig. 2b). Within the oral cavity samples throat-derived microbiomes are characterized by a decreased Shannon entropy as compared to saliva and plaque samples. In this regard, the throat samples followed a similar distribution as the stool samples. This observation posed the question of which bacterial species are driving the distances and whether those species are correlated to disease patterns. We thus computed the relative microbiome compositions at the genus level for each specimen type within each disease (Fig. 2c). The main driver for differences in microbial composition was the specimen types, prompting us to group the patterns within each type for the different diseases. Across the specimen types, our results suggested varying signatures for the most abundant bacterial genera with the competitive athletes frequently being the most deviating group. Of note, the patterns of the athletes only partially matched the standard control cohort. Especially for the skin deviating microbiome compositions are present in patients suffering from digestive tract disorders. Another obvious shift is a differential stool microbiome composition present in patients suffering from digestive tract disorders. To find significant changes in microbial compositions between disease cohorts and species, we carried out a differential abundance analysis at the species level. Splitting increased and decreased species in the different diseases confirms the complex patterns (Fig. 2d, e). For metabolic disorders and heart diseases, the largest number of significant species were recorded. Both were marked by a trend towards increased presence of species distributed across different specimen types. Among the specimen types, the oral cavity had overall the highest shares. Digestion disorders were characterized by a lower abundance of species in the stool samples at an increased frequency in the oral cavity samples. A similar pattern of higher abundance of species in the oral cavity is present in eye diseases. But these are also showing decreased species specifically in the samples taken from the eye. To get a broader overview, we grouped the patterns per specimen type and disease group (Supplementary Data 3). After adjusting $p$-values according to the number of species present in each specimen type, statistically significant differentially abundant hits emerged in nearly all cohort comparisons. Of note, the effects extended to specimen types not directly linked to the diseases; for example, 63 significant species were identified in the saliva in the case of neural disorders. Overall, for each disease group over 200 significant species were present, with saliva samples yielding the largest number of significant hits across the diseases. Remarkably, we detected a small number of significant results between controls and competitive athletes following adjustment for multiple testing. To assess the impact of comorbidity patterns on the results, we split the dataset into those patients with more than one diagnosed disease (Fig. 2f) and those with exactly one diagnosed disease (Fig. 2g). In the latter case, a reduced number of significant hits remained, suggesting that patients with comorbidities present higher alterations in the human microbiome, regardless of the body site. Because this might be partially due to the smaller number of patients with exactly one disease and the different number of samples per specimen type, we further analyzed the effect sizes as a robust measure in addition to the $p$-values. The results provided evidence for the validity of the reported patterns in general, however, also demonstrated substantial effects (absolute Cohens D $> 0.5$) in cases where the regression-based hypothesis test did not yield significance (Supplementary Fig. 3).

In order to highlight potential systematic differences between healthy and diseased cohorts, we analyzed the highest hierarchy level in our disease ontology: all patients versus all controls (Fig. 2h, Supplementary Data 4). Our results suggested that *Staphylococcus epidermidis* and *Corynebacterium pseudogenitalium* are significantly more frequent in skin swabs from patients suffering from coronary artery disease. Moreover, we detected a significant increase of *Capnocytophaga gingivalis* in saliva samples from aniridia patients, as well as *Porphyromonas endodontalis* in interdental plaque from obesity patients, and *Lachnoanaerobaculum saburreum* in interdental plaque from aniridia patients. Furthermore, *Streptococcus australis*. Lachnospiraceae bacterium oral taxon 096, Streptococcus sp. A12, Streptococcus sp. HMSC067H01, and *Aggregatibacter aphrophilus* appeared in decreased abundance in saliva in most analyzed diseases. These commensal bacteria are the five species, which across all analyzed diseases displayed a significant decrease. Only two species demonstrated differential abundance across the disease comparisons. Specifically, *Bacteroides cellulosilyticus* exhibited a significant decrease in stool samples from patients with digestive ailments, while *Streptococcus vestibularis* displayed a significant decrease in saliva samples from aniridia patients and an increase in saliva samples from Parkinson's disease patients. Overall, the abundance analysis of known and annotated bacteria yielded significant disease annotations. However, the body still seems to harbor microbes that are not characterized and annotated in databases[42]. This motivated the following analysis of assembled metagenomic data, specifically in the context of existing antimicrobial resistance (AMR) genes.

## AMR analyzes suggest pan-microbiome resistance within patients

From the 10.2 terabases of sequencing information, we generated 450 million scaffolds by computing metagenomic assemblies for each sample separately. Following the nature of short-read metagenomic sequencing, metagenomics assemblies yield short contigs. Nonetheless, 19 million fragments exceeded 1 kilobase and 300,000 fragments surpassed the 50 kb mark (Fig. 3a, Supplementary Data 5). Intriguingly, long contigs were present in samples from all specimen types, including the skin and the eye. One of the main reasons for metagenomic assemblies is to search for the presence of antimicrobial resistance genes. We thus performed resistance gene profiling based on the assembled fragments (Fig. 3b), pinpointing a consistent prevalence of the *mef(A)* resistance gene. This gene was present in 1484 of 1931 samples (77%), spanning a set of 493 of 515 distinct individuals (96%).

Overall, our data suggested that when a resistant gene is identified in one specimen of a patient, there is an increased likelihood of detecting the same resistance gene in other specimens from the same patient. (Fig. 3c). We report 120 of such significantly associated biospecimen and gene combinations after $p$-value adjustment (Fisher exact test $p$-value $< 0.05$; Supplementary Data 6). Particularly noteworthy is the significant prevalence of resistance genes observed within skin samples, which maintain continual contact with the external environment. In this context, our results suggested that the detection of a resistance gene within either the arm or forehead microbiome corresponds to a probability of 16.1% for the arm microbiome and 13.1% for the forehead microbiome to encounter the same resistance gene within the patient's stool. Furthermore, we screened for emerging resistance genes in Gram-negative bacteria against carbapenems and colistin, which constitute a global health threat. In our data, we detected several *bla-Oxa* genes, encoding various β-Lactamases in *Acinetobacter sp., Klebsiella sp., Pseudomonas sp.*, and more (Fig. 3b). Of note, we did not detect the most prevalent *bla-Oxa-48* across the study cohort. In conjunctiva swabs,

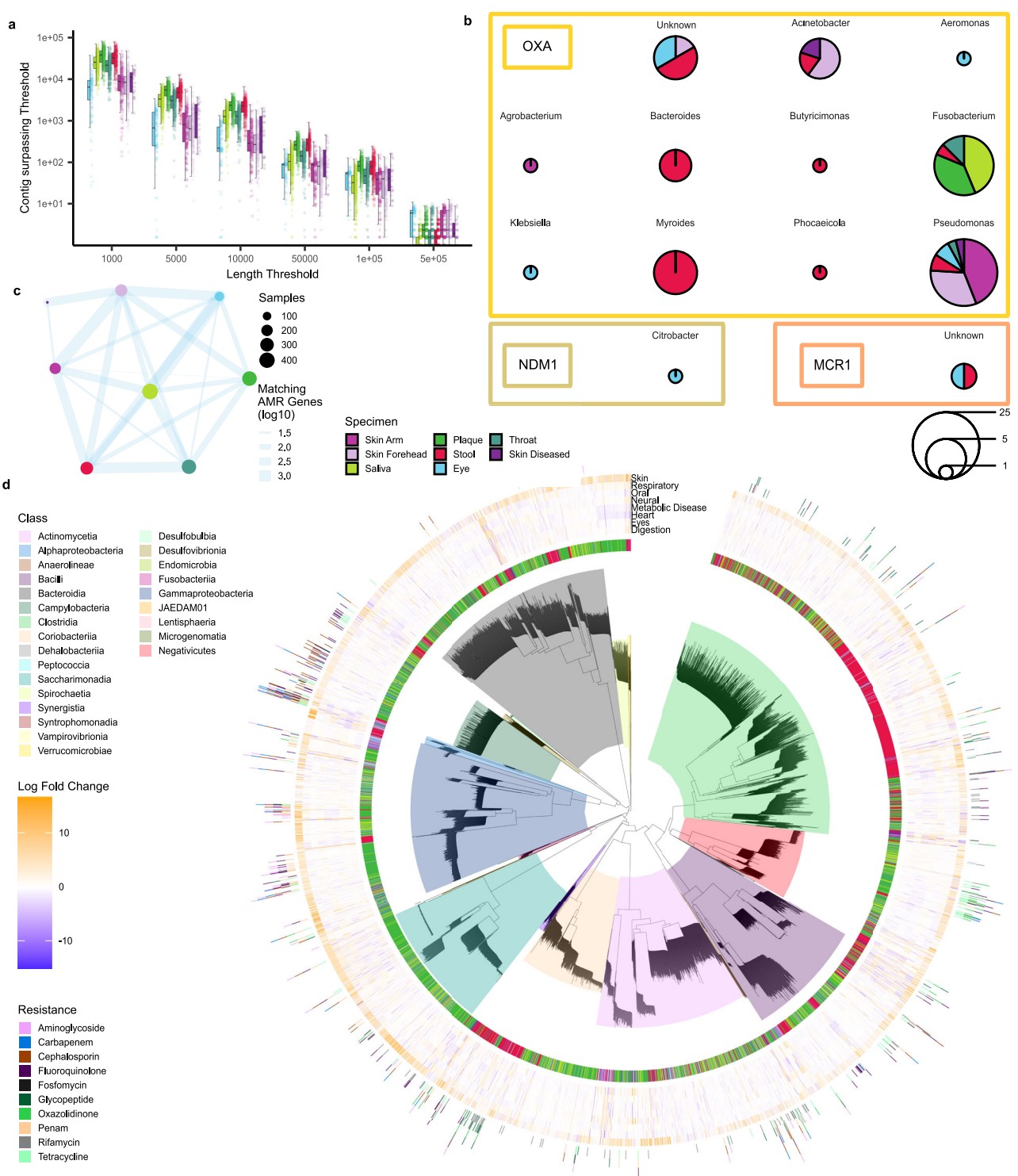

**Fig. 3 | Assembly and resistance gene analysis. a** Distribution of the number of scaffolds in each sample at various length limits, colored by specimen as box-whisker plot (n = 1931). The boxplot follows a similar style to Fig. 2h. **b** Sequence of pie charts indicating the presence of emerging antimicrobial resistance genes. Panels are subdivided by genus that was assigned to the contig where resistance genes have been detected. Pie charts scale with the number of measurements in different samples and are colored by the relative frequency of the sample's biospecimen. **c** Network visualization of counts of shared antimicrobial resistance (AMR) genes among different biospecimen samples derived from the same patient. Note, any resistance gene annotated by AMRFinderPlus was used for this plot. **d** Dereplicated SGBs defined from our data. Visualized information includes biospecimen of the initial sample where the SGB was derived from, selected resistance information taken from Pathofact, and effect size of differential coverage analysis for selected cohorts. Note, the visualized differential coverage focuses only on the biospecimen of the initial sample where the SGB has been defined from that is also visualized in the central ring.

we however observed the New Delhi metallo β-lactamase-1 (*NDM-1*) in *Citrobacter* sp. Moreover, the plasmid-mediated resistance to colistin, *mcr-1*, was found in one conjunctiva swab and one stool sample. When correlating the AMR genes to the different cohorts, we found surprisingly few statistically significant hits, suggesting that the presence of resistance genes is similar in cases and controls independent of the specimen type.

## Unveiling 314 microbial genomes associated with diseases

In light of the observed disease associations of known bacteria and the limited association patterns of AMRs in health and disease states, it is reasonable to ask for disease annotations of the not yet annotated bacteria. Therefore, we generated species genome bins (SGBs) and probed their potential links to diseases. We reconstructed a total of 4380 dereplicated microbial genomes derived from 1448 high-quality samples. SGBs were further compared to multiple references to assess their novelty, highlighting 583 undescribed SGBs (Supplementary Data 7). Of known SGBs, 146 lack species representation in GTDB r214. Out of the 4380 genomes, 886 were marked as high quality (> 90 completeness), of which 72 were displayed perfect completeness and 80 were not yet documented. Notably, oral microbiome specimens (plaque and saliva) exhibited a higher proportion of undescribed

genomes (Supplementary Fig. 4), accounting for 72% of novelty and 75% of lack of species assignments in known SGBs.

Utilizing the available coverage data, we assessed SGB enrichment within cohort-specimen combinations, revealing 10170 statistically significant combinations with an absolute log fold change exceeding two, among which 1059 involved previously undescribed SGBs (Supplementary Data 8). The pattern of disease associations in known and unannotated species, and the limited number of significant AMR genes between patients and diseases suggest other factors that might impact physiological or pathophysiological conditions in the host caused by bacteria. Here, the potential of microbes as natural producers that carry BGCs with broad functional scope must be recognized.

## Coverage-guided genome mining highlights 814 disease-related core biosynthetic genes

Among all metagenomic assembled scaffolds surpassing the 50 kb length threshold, we predicted a total of 28,315 BGCs. To identify pertinent candidates for further examination, we tailored the BigMAP[43] workflow to harmonize with our data strategy (Fig. 4a). Employing coverage profiles for each sample-predicted core biosynthetic gene pair, we acquired an informed assessment of whether a BGC exhibited enrichment or depletion within specific disease cohorts. Notably, our

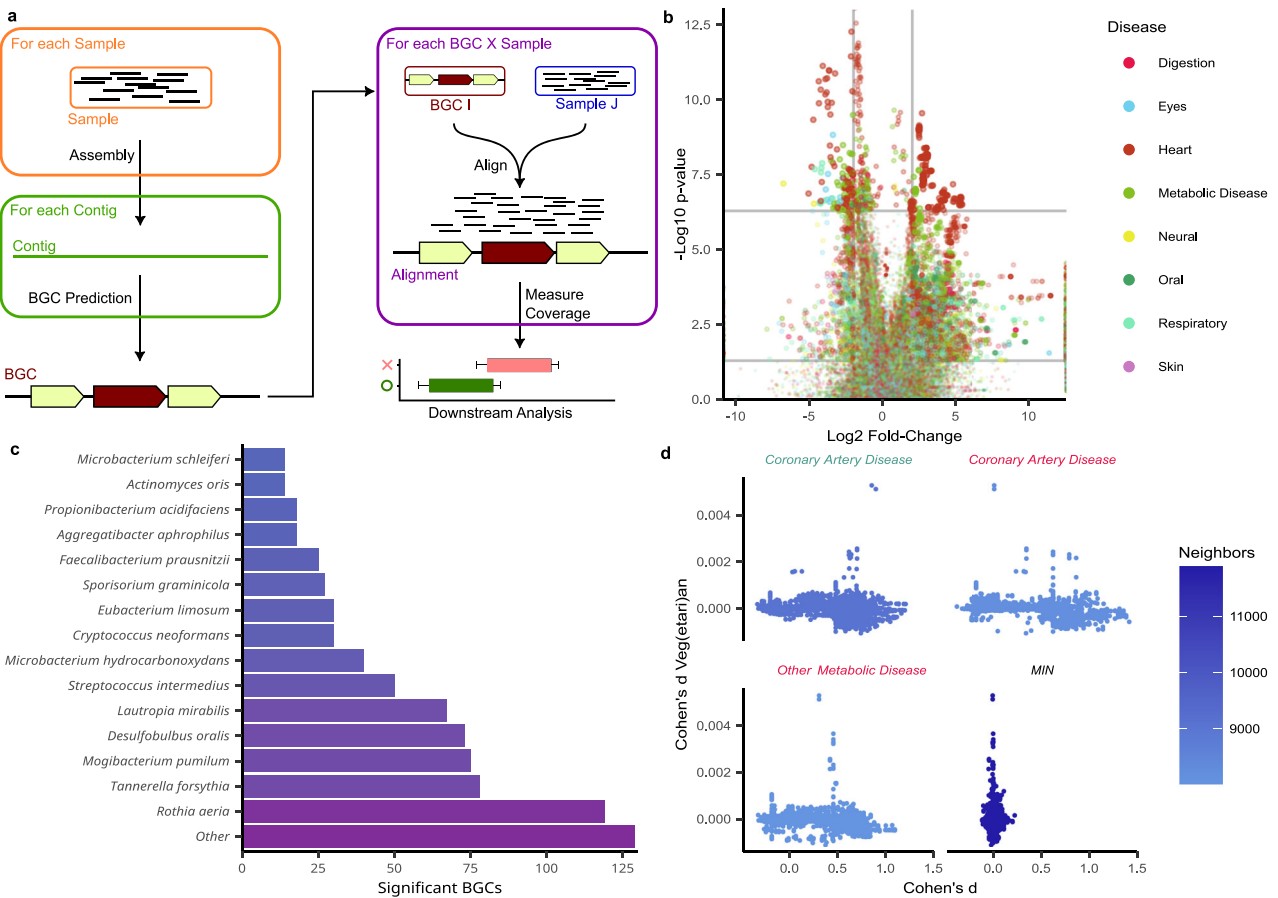

**Fig. 4 | Evidence-supported genome mining and disease association.**
**a** Schematic representation of our proposed BGC prioritization strategy representing an adapted version of the BiGMAP workflow. Metagenomic assembly is performed for each sample, followed by BGC prediction. Next, all samples are aligned against all core biosynthetic genes of predicted BGCs. Coverage information is extracted, and downstream analysis is performed. **b** Volcano plot of the differential BGC coverage analysis results. In this visualization, only matching biospecimen – initial BGC contig combinations are visualized, constituting only a fraction of all results. The unadjusted two-tailed unpaired Wilcoxon test *p*-values are shown with two horizontal lines representing the 0.05 threshold, both before

and after *p*-value adjustment. **c** Predicted host species distribution of the assembled DNA fragments where significantly associated core biosynthetic genes reside. Color reflects the number of significant BGCs. **d** Comparison of the highest correlating effect sizes, comparing differential BGC coverage results between alternative diets and diseases. The effect size of the vegetarian-omnivore comparison is visualized on the y-axis. On the x-axis, the cohort named above the panel is compared against the healthy cohort. For the fourth panel, the minimum effect size across all cohort comparisons is taken for each BGC and compared against the diet comparison.

observations reveal that 9% of core biosynthetic genes demonstrate remarkable specificity to the originating samples (Supplementary Fig. 5). Nonetheless, upon focusing exclusively on matching specimen-disease cohort pairings, we unveiled a total of 1050 statistically significant differentially altered coverages following *p*-value adjustment for the plethora of tested genes (Fig. 4b). With 119 contigs *Rothia aeria* contributed the most fragments that harbor significantly associated genes (Fig. 4c). Focusing on the functional aspect of these BGCs we observed significant differences in the distribution of BGC classes across tested cohorts, despite similar starting distributions (Supplementary Fig. 6a). In order to better understand the allegedly synthesized compounds that could be produced by the highlighted BGCs we compared them to the MIBiG database[44] (Supplementary Fig. 6c). Here, the four BGCs containing significant genes with highest similarity to existing MIBiG entries encoded for streptine with at least 80% similarity (Supplementary Fig. 6d). Consistently, all four clusters were highlighted in the same cohort. Apart from *michiganin A* no other MIBiG similarities over 40% were observed. Further exploring these findings and specifically examining individual BGCs, we searched for potential pathogenic or protective attributes, including the coverage information (i.e. whether higher or lower coverage within the control cohort was observed). It is crucial to note, however, that this methodology offers insights into the genomic presence but does not encompass the transcriptional activity of BGC genes or the overall concentration of potentially bioactive compounds. Nevertheless, the presented data strongly advocates for prioritizing future investigations into the functionality of the identified BGCs concerning their association with diseases. A prioritized search should consider features like the similarity to documented BGCs, complexity of the BGC in terms of biotechnological suitability, and quality measures linked to the prediction of the BGC.

### Impact of confounders on the microbiota showcased by the diet of individuals

As the last aspect of our study, we emphasize the importance of considering confounding factors. Inherent to study designs, such as the one applied in this study, is a broad spectrum of confounders that might impact the results. Correlations between individuals' early-life breastfeeding experience, gender, and educational attainment in relation to the microbial communities across various body sites exists[45]. Especially the sex has a large impact[46] but also factors such as ethnicity and geography[47–49]. The regional proximity and a largely shared ethnology of individuals in our study account for these factors but other obvious and non-obvious confounders remain, potentially impacting our results. One of those is the diet that impacts microbial compositions[50,51]. Because the diet was one of the variables included in our questionnaire, we performed a specific analysis of the diet, testing the dietary information related to the disease context. To this end, we also added data from a longitudinal investigation of the planetary health diet on the gut microbiome[52]. In the vegetarian stool cohort, we identified eleven significantly diminished microbial species, including those previously linked with alternative diets. Notably, species like *Bifidobacterium animals*, *Alistipes inops*, and *Phascolarctobacterium faecium*, known for producing short-chain fatty acids through dietary fiber fermentation, were more abundant in omnivorous participants. In contrast, *Dialister sp. CAG 357*, associated with inflammation, exhibited higher levels in omnivores. With respect to the SGBs, only one hit with respect to the diet remained, derived from plaque: *Saccharimonas sp013333645*. The absence of statistically significant differences in our coverage analysis might be due to the limited number of vegetarians/vegans. Nonetheless, we asked for shared signatures concerning disease correlations. Accounting for limitations in using *p*-values, we again evaluated effect sizes for each BGC and correlated them with disease effect sizes. In this analysis, we identified negative Spearman coefficients such as −0.35 for coronary artery disease in the

forehead skin microbiome, −0.30 for heart diseases in the eye, and −0.28 in the plaque of diabetes patients (Fig. 4d, Supplementary Data 9). In sum, our results provide evidence that confounding factors do have an influence on the metagenomic patterns, but despite these relevant factors, disease signals remain.

## Discussion

In our extensive metagenomic sequencing investigation, we analyzed 1931 samples following rigorous quality control. We maintained a standardized data generation protocol across multiple biospecimen samples obtained from the same individuals. Having the different microbiomes measured from the same patient offers a fairer comparison of differentially abundant microbes between different sample types and disease entities. We deliberately included and categorized a diverse spectrum of dominantly chronic inflammatory diseases, as well as globally widespread diseases. While the standardized sampling strategy and the inclusion of multiple disease cohorts represent a core strength of our study, we acknowledge the challenges that remain. One of those is the impact of obvious and less obvious confounders. The broad spectrum of diseases with different ages of onset leads to a broad age distribution. With an additional gender distribution leaning towards more males than females, with a ratio of 58.1% to 41.9% we have a second confounding factor. Others include concomitant medications, ethnicity, geographic location, and the diet. Of note, respective confounding factors are correlated to each other (e.g. the nutrition is linked to the geographic origin), making the local sampling characteristics an advantage of our study. Further, standardized operating procedures add to the stability of the results. To investigate the impact of one confounder in detail, we compared the dietary association with microbiomes in the context of disease associations with the microbiomes. These results suggest that this confounder has an impact on the metagenomes, but that the disease trajectories remained despite this influence.

One major aim of our study was to identify diagnostic patterns. Indeed, our results suggest a complex pattern of disease-to-microbiome associations depending on the specimen types. We reached the highest diagnostic power from gut and oral cavity samples. Here, the low abundance of DNA in the eye or the skin and smaller cohort sizes might lead to lower overall diagnostic values. Still, several interesting hits remained in those specimen types, especially in the case of acne inversa and the skin. It is important to highlight that all associations discovered in this study need in-depth considerations and validation. Examples of associations include the increased presence of *Otrichia sp. oral taxon 225* species in the saliva of patients suffering from Parkinson´s disease. Parkinson´s disease remains challenging to diagnose, for which additional testing for biomarkers in easily accessible body fluids, such as saliva, would provide great potential for improved diagnostic procedures[53]. However, the question of what comes first - the microbes or the disease, remains to be solved in functional studies.

Beyond diagnostic associations of microbiota to diseases, one aim of our study was the examination of antimicrobial resistances because we speculate that microbial dark matter carries resistance gene information that needs to be monitored. The most prevalent resistance gene identified across all specimens was *mef(A)*, encoding a resistance against macrolide antibiotics. This macrolide efflux gene was first described in 1996 and has emerged rapidly in *Streptococcus* sp. worldwide[54–57]. Therefore, it is not surprising that we observed such a high prevalence in our study cohort. Furthermore, we identified emerging resistance genes against carbapenem and colistin, both used to treat infections with Gram-negative bacteria. These resistances display a global health threat as treatment options are limited. The most prevalent carbapenem resistance genes are related to Oxacillin-hydrolyzing (*OXA*) carbapenemases and New Delhi metallo beta-lactamases (*NDM*)[58,59]. Colistin had been abandoned for the treatment

of Gram-negative infections for many decades but has been reintroduced as a last-resort antibiotic in the last decade. First described in 2011, the plasmid-mediated colistin resistance gene *mcr-1* displays another challenging global health threat, as it spreads rapidly and decreases the options for last-resort antibiotics in case of multi-resistant Gram-negative infections[60,61]. In our study cohort, including 515 patients from southwest Germany, we identified two patients colonized by *NDM-1* positive *Citrobacter freundii*, two patients colonized by *mcr-1* carrying *Gammaproteobacteria*, and a variety of *OXA* mediated resistances against carbapenems. Interesting are *blaOXA-50* carrying *Pseudomonas aeruginosa*, *blaOXA-270* carrying *Acinetobacter pittii*, and *blaOXA-58* carrying *Acinetobacter baumannii*. The carbapenem hydrolyzing activity of *blaOXA-50* and *blaOXA-270* has neither been confirmed nor denied. The carbapenemase *blaOXA-58*, however, was first described in 1995 and has spread globally ever since, posing one of the major carbapenem resistance genes in *Acinetobacter baumannii*[62]. We did not observe any bacterial *blaOXA-48*, which displays a now emerging resistance against carbapenems[63]. Our study setup also allows us to compare resistance genes across different specimens of the same patient. Here, resistance genes on the skin were indicative of carrying the same resistance genes in the gut.

Another important aim was to explore the functional capabilities of the measured microbiomes and to assess these abilities for potential associations to diseases. While there are many functional aspects of interest that may be explored in the future such as a vast diversity in genes, the pathways they are involved in, or the regulatory ncRNAs that may regulate them, we focused our attention on BGCs[64]. BGCs encode for molecular machineries, building natural products that are screened as a source of therapies. Our study setup was thought to enable a prioritization of BGCs with respect to therapeutic potential. By categorizing the data into distinct cohorts at various disease ontology levels, we identified BGCs that exhibited differential abundance and coverage. Beyond potential pathogenic species markers, we uncovered benign BGCs that displayed heightened coverage in healthy control groups. These BGCs warrant further exploration in vitro, offering promising avenues for medical discoveries, including the potential development of antibiotic compounds[65]. As a next step, we plan to thoroughly investigate these promising BGCs for their potential beneficial properties.

## Methods

### Clinical sampling

Clinical samples were obtained from study participants after having obtained written informed consent at Saarland University Medical Center in Homburg, Germany. Approval for the study was granted by the ethics committee of the local medical association (Ärztekammer des Saarlandes) with the identification number 131/20. Participants were not compensated. Per patient, an extensive medical examination was conducted by medical staff and diseases of interest for this study were identified. If such a disease was present, an in-depth medical history was obtained, including major factors that might influence microbial compositions in and on the human body, such as medication, lifestyle choices pertaining to diet, activity, smoking and for example alcohol uptake, as well as co-morbidities. Sex was derived from insurance documents. Subsequently, clinical samples were obtained: saliva, interdental plaque, conjunctiva swab, throat swab, stool, and skin swabs of the forehead and arm region, as well as in case of Acne inversa – affected skin areas. Concisely, fecal samples were procured from participants through utilization of a paper toilet-hat and a sterile collection tube complete with an integrated spoon, yielding an approximate range of 500 mg to 1 g of stool. Plaque samples were gathered through the use of twelve disposable micro applicators (Catalog No. MSF400, Microbrush International, Grafton, WI). Each quadrant involved brushing three interdental spaces, and subsequent transfer of all micro applicators to an ESwab transport tube (Copan Diagnostics, Brescia, Italy) along with ESwab Amies Medium (Copan Diagnostics). Saliva samples were obtained using 50-ml sterile, conic falcon tubes. Participants were instructed to deposit unstimulated saliva into the sterile falcon tube for a duration of 5 minutes. Conjunctiva specimens were acquired utilizing an ESwab. Eversion of the lower eyelid facilitated swabbing along the complete extent of the lower fornix thrice. Throat swabs were taken cautiously by medical staff, avoiding contact to saliva, tonsils, gums, and teeth, by streaking the throat-area 3–5 times. The forehead and arm were swabbed using an ESwab and wetting the nylon-flocked swab with the provided Amies Medium prior to contact with the skin. The areas were swabbed roughly to ensure removing bacterial mass not just from the surfaces, but also from hair follicles for example. Subsequent placement of the swabs into the designated transport medium was followed by freezing the tube at −80 °C.

### DNA extraction

DNA was extracted from all native samples using the Qiagen QiAamp Microbiome Kit (Qiagen, Hilden, Germany). The DNA was extracted according to the manufacturer's protocol. Briefly, swabs were vortexed in 1.5 ml Amies Medium for 2 minutes. The Amies medium containing the microbial mass from each sample was then used for DNA extraction according to the manufacturer's recommendation. For fecal samples, 250 mg of stool was used and mixed with 500 µl of buffer AHL. Micro applicators used to collect interdental plaque were mixed with 1 ml 1x PBS (pH = 7.4) and vortexed rigorously for 2 minutes. 1 ml PBS containing bacterial cells was then used for DNA extraction. Saliva samples were vortexed briefly to allow homogenization of the sample. Then, 1 ml of saliva was used for DNA extraction according to the manufacturer's recommendations. Utilizing the MP Biomedicals™ FastPrep-24™ 5 G Instrument (FisherScientific GmbH, Schwerte, Germany), mechanical disruption of bacterial cells was carried out. The operational parameters were set to a velocity of 6.5 m/s for 45 seconds, executed twice, with intervals of 5 minutes of ice storage separating each lysis cycle. After the lysis procedure, DNA was extracted into 50 µl elution buffer. To determine the DNA concentration, comprehensive microvolume UV-Vis measurements were performed using the NanoDrop 2000/2000c (ThermoFisher Scientific, Wilmington, DE)[41].

### Library Preparation, sequencing, and quality control

Extracted DNA from all native samples was sent to Novogene Company Limited (Cambridge, UK) for metagenomic library preparation and subsequent paired-end (PE150) Illumina sequencing (HiSeq). Before sequencing, potential genomic DNA degradation was measured with the fragment analyzer platform AATI (Agilent Technologies, CA, USA). The DNA concentration was measured using Qubit (Thermo Fisher, Wilmington, DE). Beads-based size selection of libraries selected for 500 bp fragments. The Novogene NGS DNA Library Prep Set (Cat No.PT004) was used for library preparation. From the study, we excluded sparsely collected biospecimens (n = 47), substandard (n = 1304), and anomalous samples (n = 201).

### Next-generation sequencing data preprocessing

The first step of data analysis was host read removal with KneadData[66] (version (v): 0.7.4; command line arguments (cla): "--trimmomatic-options = 'LEADING:3 TRAILING:3 MINLEN:50' --bowtie2-options = '--very-sensitive --no-discordant --reorder'"). Due to the high contamination load among skin and eye samples, we additionally ran the human sra-human-scrubber[67] (v: 1.0.2021_05_05) after KneadData. Paired-end reads were only kept if none of the read pairs mapped to the human reference. After decontamination, we performed sequence overrepresentation analysis and quality assurance with fastp[68] (v: 0.20.1; cla: "--overrepresentation_analysis") and visualized results with MultiQC[69] (v: 1.11).

## Reference-free beta-diversity analysis

Mash[70] (v: 2.3; cla: "sketch -S 23 -k 31 -s 5000 -r -m 2") was used to compute and compare MinHash distances. A two-dimensional embedding was generated in R with the UMAP package[71] (v: 0.2.8). After noticing the separation of the low-input biospecimen into two clusters, we split the low-input samples by their clustering behavior during outlier removal. During this outlier removal, we performed for each biospecimen a Grubb's test on the mean of all pairwise MinHash distances and removed the most significant outlier. This procedure was repeated iteratively until no more significant outliers were left. In the case of low-input biospecimen, this algorithm was performed for each subcluster instead.

## Reference-based compositional analysis

MetaPhlAn3[66] (v: 3.0.13; cla: "-t rel_ab_w_read_stats --unknown_estimation --add_viruses") on the mpa_v30_CHOCOPhlAn_201901 database was used to profile quality controlled samples. Relative counts were rescaled to absolute counts based on the number of reads and virus counts were removed. Shannon diversity was used as an alpha-diversity measure. Reference-based beta-diversity was assessed with non-metric multidimensional scaling on Bray-Curtis distances. Differential abundance analysis was performed with ANCOM-BC[72] (v: 1.6.2). P-values were adjusted via Benjamini-Hochberg adjustment. Note, that we only tested specimen-cohort combinations with more than ten samples in each cohort. While the athletic and sports cohorts were tested against healthy controls, all other diseases were tested against the union of healthy control and sports cohorts, i.e. the healthy cohort. Samples that were part of the control and disease cohort during testing, such as athletes with diseases, were removed. Based on the differential abundance analysis results, we searched for interesting pathogens and commensal bacteria to further investigate. To this end, we defined the pathogenicity score as the number of various diseases where a pathogen is predicted to be significantly more abundant in the diseased cohort in at least one biospecimen. Similarly, we define the commensal score as the number of disease cohorts where a bacterial species is significantly reduced in the diseased cohort for at least one biospecimen. In visualizations of abundances, absolute counts were center log ratio normalized.

## Metagenomic assembly

We assembled each sample with SPAdes[73] (v: 3.15.4; cla: "--meta") and monitored the assembly quality with QUAST[74] (v: 5.0.2; cla: "-s"). Scaffolds were binned with MetaBAT2[75] (v: 2.15; cla: "--seed 420"). MAGs across all samples were aggregated and dereplicated with dRep[76] (v: 3.4.2; cla: "-comp 50 -con 5 --checkM_method lineage_wf --S_algorithm fastANI --S_ani 0.95 -nc 0.5"). Mid and high-quality genomes were selected for subsequent analysis using a similar MIMAG's[77]: mid-quality genomes (MQ) with completeness >= 50% and contamination <5%, and completeness >90% and contamination <5% for high-quality (HQ) genomes. MAGS with average differences <5% ANI were clustered into 4,380 SGBs. To assess the novelty of the SGBs, we conducted a comparison against multiples references (99,376 genomes), including GTDB r214[78], the Unified Human Gastrointestinal Genome collection[79], the Singapore Platinum Metagenomes Project[80], and Pasolli et al.[81], utilizing Mash distances (<= 0.05) followed by validation through FastANI[82]. Based on the comparison, SGBs were classified into two categories: (i) known, if a match to a known source was found (>= 95% ANI), (ii) novel, if no matches were found. If multiple matches were found for a sequence, the match with the best ANI was retained. GTDB-TK[83] (v: 2.3.0; cla: "classify_wf") in combination with GTDB r214[78] was used to annotate SGBs with taxonomic information. Differential coverage analysis on SGBs was performed by first, aggregating all bins creating one reference file, and then aligning all samples against the newly created reference file with Bowtie2[84] (v: 2.3.4.3; cla: "-a"). Afterward, coverage information

was extracted from each alignment with SAMtools idxstats[85] (v: "1.16.1"). Differential coverage analysis was organized identically to the differential abundance analysis, i.e. cohort-specimen multiplicity needed to be larger than ten and the control cohorts altered if sports or athletes were considered for testing. However, here, a Wilcoxon rank sum test was executed, performing Benjamini-Hochberg adjustment to adjust for the total number of tested dereplicated SGBs.

## Virulence Analysis

On each dereplicated SGB, we ran PathoFact[86] (commit v: 55d8240). To capture resistances that did not end up in any final bins, we ran AMRFinderPlus[87] (v: 3.11.4) and complemented the information with Kraken[88] (v:2.1.2; cla: "--use-mpa-style"; database version: k2-pluspf_20220908) taxonomic classifications for each contig.

## Genome mining

After assembly, all contigs with a length > 50,000 bp were mined for BGCs with antiSMASH[89] (v: 6.1.1; cla: "--genefinding-tool prodigal --cb-knownclusters --cb-subclusters --asf"). Next, all core biosynthetic genes that were annotated by antismash were extracted and aggregated into one file. Reads of all samples were then aligned against this reference and coverage information was extracted for each contig-sample combination, following the procedure described for the SGB analysis. Similarly, differential coverage analysis was repeated identically to the SGB analysis. P-value adjustment was performed using Benjamini-Hochberg adjustment, adjusting for the number of tested core biosynthetic genes. Taxonomic assignment of BGC fragments was performed using Kraken2[88] (v: 2.1.2; cla: "--use-mpa-style") with the standard database (v:k2_pluspf_20220908).

## Dietary alterations comparison

All day zero samples from the dataset of Rehner et al.[52] were taken and processed according to the new data. The dataset was then extended by our dataset, however, only using our healthy controls. The dataset was from then on analyzed independently. If comparisons were performed, e.g. in statistical tests, the vegan/vegetarian cohort was always compared to the omnivore cohort.

## Reporting summary

Further information on research design is available in the Nature Portfolio Reporting Summary linked to this article.

## Data availability

The metagenomic sequencing data after removing ambient human DNA generated in this study has been deposited in the Sequencing Read Archive under the accession code PRJNA1057503.

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

## Acknowledgements

Our appreciation extends to all individuals associated with Saarland University, Saarland University Medical Center, and Helmholtz Institute for Pharmaceutical Research Saarbrücken, whose contributions significantly enriched this study, albeit not reflected in authorship. Furthermore, we extend our gratitude to the participants of this study, whose consent allowed us to acquire and scrutinize clinical samples, thereby unraveling insights into their intricate microbial composition and functions. The compute infrastructure for this project was funded by the DFG [469073465], received by AK. This work was further supported financially by the Saarland University, the UdS-HIPS TANDEM initiative, and the TALENTS Marie Skłodowska-Curie COFUND-Action of the European Commission [101081463], received by LAGM. The views and opinions expressed are, however, those of the authors only and do not necessarily reflect those of the European Union, which cannot be held responsible for them.

## Author contributions

R.M., S.L.B., A.K., and R.B. supervised the research and obtained research funding. S.L.B., A.K., G.P.S., and J.R. designed the experiments.

J.R. and N.L. performed the experiments, and together with GPS and LAGM analyzed the data. G.P.S., A.G., and L.A.G.M. designed the computational analysis. M.H., B.S., S.K., F.M., V.K., M.K., T.M., T.B., E.F., B.S., S.F., S.S.G., S.K., M.Z., F.G., Jö.R., T.V., C.H., M.S., M.L.S., M.U., A.B., N.L.H., M.C.M., J.M.S., and R.B. supervised clinical sample collection and medical assessment of participants and contributed to the medical and experimental workflow. M.P.G., S.R., T.B., E.F., S.F., S.S.G., M.K., F.G., V.K., C.G., L.D., and A.H. collected clinical samples and performed medical assessment of participants health. J.R., G.P.S., and A.K. wrote the paper. All authors reviewed and edited the paper. D.K., J.H., K.B., T.A.M.G., C.F. C.B., and O.V.K. supported the annotation of the Biosynthetic Gene Clusters. J.M.S. supported the interpretation of the gut-related microbiota.

## Funding

## Competing interests

G.P.S., R.M., and A.K. are co-founders of MooH GmbH, a company developing metagenomic based oral health tests. FM is supported by Deutsche Gesellschaft für Kardiologie (DGK), Deutsche Forschungsgemeinschaft (SFB TRR219, Project-ID 322900939), and Deutsche Herzstiftung. His institution (Saarland University) has received scientific support from Ablative Solutions, Medtronic, and ReCor Medical. He has received speaker honoraria/consulting fees from Ablative Solutions, Amgen, Astra-Zeneca, Bayer, Boehringer Ingelheim, Inari, Medtronic, Merck, ReCor Medical, Servier, and Terumo. The remaining authors declare no competing interests.

## Additional information

Georges P. Schmartz[1,18], Jacqueline Rehner[2,18], Madline P. Gund[3,18], Verena Keller[4,18], Leidy-Alejandra G. Molano[1], Stefan Rupf[3,5], Matthias Hannig[3], Tim Berger[6], Elias Flockerzi[6], Berthold Seitz[6], Sara Fleser[7], Sabina Schmitt-Grohé[7], Sandra Kalefack[7], Michael Zemlin[7], Michael Kunz[8], Felix Götzinger[8], Caroline Gevaerd[9], Thomas Vogt[9], Jörg Reichrath[9], Lisa Diehl[1], Anne Hecksteden[10,11], Tim Meyer[10], Christian Herr[12], Alexey Gurevich[13,14], Daniel Krug[13], Julian Hegemann[13,15], Kenan Bozhueyuek[13], Tobias A. M. Gulder[13,15], Chengzhang Fu[13], Christine Beemelmanns[13], Jörn M. Schattenberg[4], Olga V. Kalinina[13], Anouck Becker[16], Marcus Unger[16], Nicole Ludwig[1], Martina Seibert[6], Marie-Louise Stein[6], Nikolas Loka Hanna[12], Marie-Christin Martin[6], Felix Mahfoud[8], Marcin Krawczyk[4], Sören L. Becker[2,18], Rolf Müller[13,17,18], Robert Bals[12,17,18] & Andreas Keller[1,13,17,18] ✉

[1]Clinical Bioinformatics, Saarland University, 66123 Saarbrücken, Germany. [2]Institute of Medical Microbiology and Hygiene, Saarland University, 66421 Homburg, Germany. [3]Clinic of Operative Dentistry, Periodontology and Preventive Dentistry, Saarland University, 66421 Homburg, Germany. [4]Department of Medicine II, Saarland University Medical Center, 66421 Homburg, Germany. [5]Synoptic Dentistry, Saarland University, 66421 Homburg, Germany. [6]Department of Ophthalmology, Saarland University Medical Center, 66421 Homburg, Germany. [7]Department of General Pediatrics and Neonatology, Saarland University, 66421 Homburg, Germany. [8]Department of Internal Medicine III, Cardiology, Angiology, Intensive Care Medicine, Saarland University Hospital, 66421 Homburg, Germany. [9]Clinic for Dermatology, Venereology, and Allergology, 66421 Homburg, Germany. [10]Institute for Sport and Preventive Medicine, Saarland University, 66123 Saarbrücken, Germany. [11]Chair of Sports Medicine, Institute of Physiology, Medical University of Innsbruck, Innsbruck, Austria. [12]Department of Internal Medicine V - Pulmonology, Allergology, Intensive Care Medicine, Saarland University, Saarbrücken, Germany. [13]Helmholtz Institute for Pharmaceutical Research Saarland, 66123 Saarbrücken, Germany. [14]Center for Bioinformatics Saar and Saarland University, Saarland Informatics Campus, 66123 Saarbrücken, Germany. [15]Department of Pharmacy, Saarland University, 66123 Saarbrücken, Germany. [16]Department for Neurology, Saarland University Medical Center, 66421 Homburg, Germany. [17]PharmaScienceHub, 66123 Saarbrücken, Germany. [18]These authors contributed equally: Georges P. Schmartz, Jacqueline Rehner, Madline P. Gund, Verena Keller, Sören L. Becker, Rolf Müller, Robert Bals, Andreas Keller. ✉ e-mail: andreas.keller@ccb.uni-saarland.de

