## [Peer Review File · Nature Communications]

Decoding the diagnostic and therapeutic potential of microbiota using pan-body pan-disease microbiomicsReviewers' Comments:

Reviewer #1:

Remarks to the Author:

The paper by Schmartz, Rehner et al. presents a rich catalogue of metagenome sequences, metagenome-assembled genomes and biosynthetic gene clusters from a very large number of clinical samples. The dataset is of high quality and is among the best and largest datasets of clinical microbiome samples ever collected.

While few specific biological conclusions can yet be drawn from the data presented, the study provides a strong basis for follow-up research as a key resource for microbiome science.

I have a number of specific comments that the authors may use to their advantage:

- Lines 301-315: how many of the SGBs are novel compared to those previously reported in human microbiome bin catalogues? Are any associations with diseases in line different from (or in line with) what has been reported previously? Some more contextualization with literature would be helpful.
- Lines 318-335: While BGCs are definitely interesting, it would be useful to clarify why these were the singular focus of the functional annotation efforts. If analysis of other metabolic pathways and genomic traits is not performed in this paper, it could at least be mentioned as a prospect for future studies in the Discussion.
- Line 323: can you be more specific here? How many BGCs are sample-specific? Can a distribution be plotted and included as SI Figure?
- Lines 318-335: were any known BGCs or BGCs related to those with known products identified? If so, did they show any meaningful patterns across samples with different clinical metadata? Were BGCs associated with specific compound classes enriched with regard to any disease phenotypes?
- Line 432: the prioritization of BGCs is mentioned, but it seems the data resource still presents many more potentially relevant BGCs than can be experimentally characterized. Can the authors discuss in a bit more depth how they think this prioritization should be carried out?
- Figure 4C: to me it is unclear what the message of this figure panel is and why it is relevant.

Minor:

- Line 95: reference missing?
- Lines 255-256: strain identifiers should not be italicized

Reviewer #2:

Remarks to the Author:

Microbiome of various body sites has the potential to be linked to etiology of many human diseases and complications. How bacteria influence homeostasis at local or distant sites had remained unclear for the majority of diseases and contexts. Here, authors measure the microbiomes of 657

individuals, generating 3483 samples from oral cavity, skin, throat, stool and eye. A particular strength of this study is the capacity to compare samples from the same individual, which have on average measurements from at least 3 body sites.

For the most part, the results highlight the scale and size of the data collected, and state claims without sufficient analysis or experimental support.

The datasets could indeed present an important "resource for future investigations", however the time and place for some of these investigations would be right here. In particular, functional implications of the associations are missing from all of the analysis, leaving the reader without a single actionable task. Should some species be therapeutically targeted? Do any identified BGC suggest a causative role in any one disease? Do AMR or BGCs improve fitness of healthy/disease associated bacteria?

One motivation of the study was to "define a healthy microbiome". This is an ambitious task and has been tackled by many further studies without clear conclusions (PMID 31142855, 29255284, 27122046). Due to extremely high diversity of microbiomes it is unclear how such an endpoint would be defined, or whether it has an unequivocal answer. While this dataset excels in diversity of diseases hypothetically linked to microbiome, only five species in saliva were found consistently decreased across diseases. To label them as a "healthy microbiome" would be an overstatement in absence of detailed functional analyses and comparisons in independent cohorts of similar design.

The resistance gene analyses advocates that AMR genes persist within different individuals. This observation is not particularly surprising, since it has been observed that microbiomes are individualized and resilient (PMID 31194939, 33440171), and it would be helpful to illustrate whether AMR genes stand out in particular.

Deeming the 729 species level bins as "new" also does not sit right. The lack of mapping to known species always happens, most likely due to lack of coverage to explain all scaffolds, incomplete catalogs, etc. To label them "new" would necessarily include comparison to other cohorts of similar design.

Using biosynthetic gene cluster predictions, the authors identify >28,000 BGC present in the metagenomics datasets, with ~1,000 showing statistically significant signals with disease. The analysis then abruptly stops and any further information about these BGCs is missing: which species encode them, what are the BGCs families and are there any predicted molecular products. Do they provide further implications for community structures? The authors fail to mention a landmark study seeking BGCs in human metagenomics datasets (PMID: 25215495) that provides a blueprint for such investigations. The authors deem that "presented data strongly advocate for prioritizing future investigations", with no guidelines or examples of signals warranting such investments.

In summary, we deem the current version of the manuscript premature for publishing. However, we

are encouraged by careful processing and quality controls that promise identification of species and pathways strongly predicting the mentioned conditions.

REVIEWER COMMENTS

Reviewer #1 (Remarks to the Author):

The paper by Schmartz, Rehner et al. presents a rich catalogue of metagenome sequences, metagenome-assembled genomes and biosynthetic gene clusters from a very large number of clinical samples. The dataset is of high quality and is among the best and largest datasets of clinical microbiome samples ever collected.

While few specific biological conclusions can yet be drawn from the data presented, the study provides a strong basis for follow-up research as a key resource for microbiome science.

Reviewer 1 Comment 1

We appreciate the reviewer's recognition of the importance of the dataset we have presented. This collection represents a substantial repertoire of metagenome sequences, metagenome-assembled genomes, and biosynthetic gene clusters derived from a diverse array of clinical samples. We agree that specific biological conclusions are not yet drawn from the data. As we emphasize in the revision even better, the intention of our work was to lay the basis for such analyses. We believe that the study lays a solid foundation for such further research that we and others already initiated. Based on the feedback from both reviewers we added analyses to the revised manuscript that should facilitate this task even more. We are confident that the research community will derive great value from this high-quality dataset and its description due to its scope and depth.

I have a number of specific comments that the authors may use to their advantage:

- Lines 301-315: how many of the SGBs are novel compared to those previously reported in human microbiome bin catalogues? Are any associations with diseases in line different from (or in line with) what has been reported previously? Some more contextualization with literature would be helpful.

Reviewer 1 Comment 2

A total of 4,380 species-level genome bins (SGBs) were reconstructed after being submitted to strict quality control and clustering as described in the *Methods* section. Despite the

sequencing efforts made in the last years¹⁻⁴, 583 SGBs were found novel (**Supplemental Fig. 4a**), 80 classified as high-quality (>90 completeness, <5% contamination), four of which displaying 100% completeness. The higher representation of known SGBs was found in the GTDB database (54%), followed by Pasolli et al. 2019 (24%), UHGG (13%), and SPMP (8.4%) (**Supplemental Fig. 4b**) (**Supplemental Table S7**). Of known SGBs, 146 lack species representation in GTDB r214. Notably, oral specimens presented a higher rate of novel genomes and proportion of unknown species, accounting for 87% of novelty and 75% of species lack representation.

We also used this new annotation to further enrich the differential coverage analysis between SGBs and diseases groups (**Supplemental Fig. 4c**). As stated in the manuscript, 10,170 significant combinations ($|\logFC| > 2$ & $p\text{-value} < 0.05$) were found between cohorts and SGBs, among which 1,059 involved novel SGBs. Out of 1,625 SGBs (189 novels) associations with diseases, 886 (54%) were exclusively linked to a single disease group, 628 (39%) were linked to two or three disease groups, and 111 (7%) were linked to four or more disease groups. Heart disease group had the highest number of significant associated SGBs ($|\logFC| > 2$ & $p\text{-value} < 0.05$) and a higher rate of SGBs exclusively linked to a single disease group (60%). Similarly, the oral and metabolic disease groups also showed a substantial number of significant associations, consistent with the trend observed in the differential abundance analysis at species level between disease cohorts. Overall, these findings highlight that our dataset is a powerful resource, not only valuable for further exploration of known microbial species potentially linked to diseases, but also for the future identification and characterization of new microbes, especially in the context of cardiovascular, metabolic, and oral diseases. We thank the reviewer for pointing out this missing analysis and we integrated major portions of this response into the main manuscript which can be found as updated amounts of SGBs we detected in the abstract, in the manuscript results section line 311 and the following, as well as in the methods section line 516ff.

Supplemental Figure 4: a) Total number of our dereplicated SGBs visualized by specimen of the initial sample and novelty. b) Overlap of dereplicated SGBs with the Genome Taxonomy Database (GTDB), a study by Pasolli et al., the Singapore Platinum Metagenomes Project (SPMP), and the Unified Human Gastrointestinal Genome (UHGG) collection. Note, this plot is not adjusted for the size of the reference database. c) Significant SGB-disease associations computed based on coverage information. NAs indicate a lack of assignment.

- Lines 318-335: While BGCs are definitely interesting, it would be useful to clarify why these were the singular focus of the functional annotation efforts. If analysis of other metabolic pathways and genomic traits is not performed in this paper, it could at least be mentioned as a prospect for future studies in the Discussion.

Reviewer 1 Comment 3

We appreciate the reviewer's insightful perspective on the matter. Indeed, there are numerous annotations that can be performed on metagenomic sequencing data, including assessments of metabolic pathway abundance and individual gene classes using tools like HUMAnN3⁵.

Our decision to focus on biosynthetic gene clusters (BGCs) stems from the composition of our research network and the role of this project within a larger research scope. Our ultimate goal is to further assess selected BGCs, express them, describe the molecules they synthesize, and characterize their properties. However, these downstream experiments are both time-consuming and costly, which justifies our extensive attention dedicated to a thorough analysis of BGCs in order to prescreen for relevant candidates.

We acknowledge that from a reader's perspective, without this background knowledge, it may seem unusual to focus so much on BGCs without further evaluation of other functional properties. As suggested by the reviewer, we briefly elaborated on the potential for further analysis in the discussion section line 452ff, also motivating fellow researchers with expertise in the respective areas to utilize the power of the data set. In the manuscript we state:

Another important aim was to explore the functional capabilities of the measured microbiomes and to assess these abilities for potential associations to diseases. While there are many functional aspects of interest that may be explored in the future such as a vast diversity in genes, the pathways they are involved in, or the regulatory ncRNAs that may regulate them, we focused our attention on BGCs.

- Line 323: can you be more specific here? How many BGCs are sample-specific? Can a distribution be plotted and included as SI Figure?

Reviewer 1 Comment 4

We consider a BGC to be sample specific if we the BGC is only found in the sample that it has been initially discovered from. Since BGCs are highly variable in length, gene composition and similarity, we decided to reason based only on the core biosynthetic genes that are arguably most defining for a BGC. Similarly, we always consider the median coverage of these genes in order to reduce the effects of similarity among core biosynthetic genes in potential repetitive regions. Following this definition a total of 28109 (9%) core biosynthetic genes, displayed a median coverage of zero in all samples. Of course, aligning the samples that were used to predict these genes in the matching BGCs yielded non-zero

median coverage. Concerning the distribution of median coverages, we visualized the results in **Supplemental Figure 5**. Here this sample specificity is clearly visible as well. Indeed, we deemed this result interesting enough to mention it very shortly in the initial manuscript. In hindsight, we agree with the reviewer that if we mention this observation, we should also provide the supporting data for the reader. Accordingly, we added additional information in the main manuscript and added the figure as a supplemental material. We thank the reviewer for bringing this previous shortcoming to our attention and changed it accordingly in the results section of our manuscript, line 335.

Supplemental Figure 5: Graphical representation of BGC specificity. The different panels indicate the specimen of the initial sample where the initial BGC prediction derived from. Colors indicate the specimen of the aligned samples. Reads were aligned against each core biosynthetic gene in the BGCs and median coverages were computed for each sample – gene pair.

- Lines 318-335: were any known BGCs or BGCs related to those with known products identified? If so, did they show any meaningful patterns across samples with different clinical metadata? Were BGCs associated with specific compound classes enriched with regard to any disease phenotypes?

Reviewer 1 Comment 5

Using antiSMASH, we compared our in-silico predicted BGCs to MIBIG, a curated database of confirmed BGCs. Measuring similarity between BGCs, as well as predicting similarity of the resulting compounds is very difficult and subject to extensive research. For the sake of

this answer, we limit ourselves to the BLAST similarity score which is provided by antiSMASH. The score designates the relative amount of genes in a reference BGC that could be found in the query BGC. After extracting this information from the antiSMASH reports, we identified 7,299 BGCs that produce known products at varying degrees of similarity (**Supplemental Fig. 6c**). If we only focus on the 1050 significant core biosynthetic genes – disease associations mentioned in the manuscript, we observed that a total of 178 similarities larger than zero were found among the 814 different BGCs. Subsetting on reliable similarities larger than 50% we see four clusters with at least 80% similarity to the lanthipeptide streptin that differ significantly in people with heart diseases. We visualized the coverage profiles for these clusters in **Supplemental Fig. 6d**. Before analyzing if specific compound classes were enriched with regard to any disease phenotypes, we first provide an overlook of the overall amount of predicted BGC classes in **Supplemental Fig. 6a** and observe all relatively equal distributions of predicted across specimen.

If we look at the specific compound class distribution of the 1050 significant associations, we do observe significant unequal distributions of classes (X^2 -test p-value $<2.2*10^{-16}$, **Supplemental Fig. 6b**). Note that the sum of visualized core biosynthetic gene cluster exceeds 1050 as one core biosynthetic gene can have multiple annotations and count towards several different BGCs, which can be found in the results section line 339ff.

Supplemental Figure 6: **a)** Classes of the predicted BGCs divided by cohort of the initial sample. The total counts of predicted classes in the cohort are specified next to each row. **b)** Same as a) but focusing only on BGCs containing a significantly associated biosynthetic gene. **c)** Highest predicted non-zero similarity for all BGCs containing a significantly associated biosynthetic gene. The comparison was made against MIBiG. **d)** Median coverage distribution across tested samples for four highlighted BGC clusters that displayed a high similarity to streptin.

- Line 432: the prioritization of BGCs is mentioned, but it seems the data resource still presents many more potentially relevant BGCs than can be experimentally characterized. Can the authors discuss in a bit more depth how they think this prioritization should be carried out?

Reviewer 1 Comment 6

We thank the reviewer for their interest in this research question. Prioritization may leverage several features that can be associated with the different BGCs. Apart from general sample or assembly quality control measures of the fragment, the following properties may be considered before further experimental assessment:

- BGC class, length, and complexity
- Predicted host species of the fragment
- Similarity to known BGCs in databases like MIBiG⁶, BiG-FAM⁷, ABC-HuM⁸, or antiSMASH-db⁹
- Whether the BGCs are completely or incompletely located on their fragments
- Coverage distribution of the original sample
- Coverage signals across the different cohorts

We decided to first prioritize based on cohort coverages. Among these relevant candidates, we then try to select the most promising BGCs taking all other features into account. For the readers, we added several of these selection features into the manuscript text in line 355.

- Figure 4C: to me it is unclear what the message of this figure panel is and why it is relevant.

Reviewer 1 Comment 7

The main message of the panel was to provide a few examples of the different results in the volcano plot Figure 4B. Nevertheless, we agree with the reviewer that the overall relevance of the Figure is not high enough to take up valuable space in Figure 4. Therefore, we substituted it by an analysis suggested by Reviewer 2. We thank the reviewer for voicing their concern.

Minor:

- Line 95: reference missing?
- Lines 255-256: strain identifiers should not be italicized

Reviewer 1 Comment 8

We added the missing reference. Further, we removed the italicization. We thank the reviewer for spotting and highlighting these imperfections.

Reviewer #2 (Remarks to the Author):

Microbiome of various body sites has the potential to be linked to etiology of many human diseases and complications. How bacteria influence homeostasis at local or distant sites had remained unclear for the majority of diseases and contexts. Here, authors measure the microbiomes of 657 individuals, generating 3483 samples from oral cavity, skin, throat,

stool and eye. A particular strength of this study is the capacity to compare samples from the same individual, which have on average measurements from at least 3 body sites.

For the most part, the results highlight the scale and size of the data collected, and state claims without sufficient analysis or experimental support.

The datasets could indeed present an important "resource for future investigations", however the time and place for some of these investigations would be right here. In particular, functional implications of the associations are missing from all of the analysis, leaving the reader without a single actionable task. Should some species be therapeutically targeted? Do any identified BGC suggest a causative role in any one disease? Do AMR or BGCs improve fitness of healthy/disease associated bacteria?

Reviewer 2 Comment 1

We appreciate the reviewer's thoughtful comments regarding our manuscript. Indeed, highlighting the scale, size, and also the quality of the data was instrumental for us. Especially for later analyses, e.g. using language models, by the community such considerations are important, also contributing to the FAIR data principles.

We acknowledge that our analysis primarily focuses on statistical associations rather than providing causal or functional insights into the microbiome, and would like to take the opportunity clarifying some general aspects. The comment of this reviewer made us check the respective aspects and messages through the revision, especially we want to avoid making claims that are not supported by data or analyses.

We believe that our work does offer actionable tasks for readers, as highlighted by the reviewer. For instance, we provide candidate species that could serve as potential therapeutic targets, as outlined in **Supplementary Table 4**. Similarly, we identify BGCs with significantly different coverages across cohorts, which serve as valuable starting points for further investigation and experimentation, as partly voiced by the reviewer themselves.

Furthermore, in the context of the scope of our work, the depth and timing was intentionally selected to stimulate follow up work, which is beyond the scope of a first description of the IMAGINE project that collected data and samples, performed substantial sequencing and computational analyses already for over five years. We reached this aim: our research has already sparked additional initiatives within our network, and we have concurrent ongoing efforts towards experimental validation of BGC candidates. We understand that scientific

progress can be slow and resource-intensive. Thus, we believe that sharing our data, analysis, methodology, and insights with the research community is a crucial step in advancing the field. Moreover, we contend that the depth of our analysis surpasses many comparable initiatives that have been published.

We also acknowledge that there are limitations to our study. We are committed to contributing to the advancement of microbiome research and will continue to refine our approach based on constructive criticism.

One motivation of the study was to "define a healthy microbiome". This is an ambitious task and has been tackled by many further studies without clear conclusions (PMID 31142855, 29255284, 27122046). Due to extremely high diversity of microbiomes it is unclear how such an endpoint would be defined, or whether it has an unequivocal answer. While this dataset excels in diversity of diseases hypothetically linked to microbiome, only five species in saliva were found consistently decreased across diseases. To label them as a "healthy microbiome" would be an overstatement in absence of detailed functional analyses and comparisons in independent cohorts of similar design.

Reviewer 2 Comment 3

We appreciate the thoughtful insights provided by the reviewer regarding the framing of our study's motivation. Upon reflection, we acknowledge that our initial phrasing may have conveyed an overly ambitious goal. Rather than seeking to definitively define the healthy microbiome, our objective is to delineate the boundaries of microbiome compositions associated with healthy individuals compared to a wide range of different diseases, while mitigating potential biases in our data inherent through the aggregation into one large experiment.

Furthermore, we recognize the complexity inherent in characterizing a singular "healthy microbiome" due to the vast diversity observed across populations and individuals. Our study aims to address this challenge by identifying consistent microbial patterns indicative of health across diverse disease states. However, we concur with the reviewer that labeling these findings as representative of a universally "healthy microbiome" would be premature without comprehensive functional analyses and validation in independent cohorts with similar study designs.

In response to the reviewer's concerns, we would like to highlight that our research has already spurred the initiation of three follow-up studies. One of these follow-up studies is

specifically dedicated to exploring the presence and role of benevolent microbial community members within the oral microbiome. We anticipate that these subsequent investigations will provide valuable insights into the complex dynamics of microbial communities and their implications for human health. Nevertheless, we also believe that sharing our best leads and signals aiming to investigate this research question is of interest for the community and therefore would continue to include the result.

In response to the reviewer's feedback, we changed the section in the text, line 251, that was referenced by the reviewer to:

In order to highlight potential systematic differences between healthy and diseased cohorts, we analyzed the highest hierarchy level in our disease ontology: all patients versus all controls.

Further, we included the references mentioned by the reviewer to further put our research into context (line 61). We hope that these modifications we made in the main manuscript are to the satisfaction of the reviewer.

The resistance gene analyses advocates that AMR genes persist within different individuals. This observation is not particularly surprising, since it has been observed that microbiomes are individualized and resilient (PMID 31194939, 33440171), and it would be helpful to illustrate whether AMR genes stand out in particular.

Reviewer 2 Comment 4

We thank the reviewer for their comment on the antimicrobial resistance analysis we performed on all samples. We agree that it is already known that AMR genes can be detected in e.g. the human intestine in different samples in longitudinal studies. However, here we highlight that if we can detect AMR genes in one body compartment, the likelihood of detecting the same gene across different body sites in the same individual is increased. As the AMR genes detected were specific for each individual, it would be difficult to generalize one AMR gene that stood out during our analysis. We therefore highlighted the most prevalent AMR gene across all samples, *mefA*, and further investigated the presence of emerging resistances against carbapenems in Gram negative bacteria, such as OXA-48, NDM-1, and KPC-2 (line 442ff). Monitoring carbapenem resistances is crucial because carbapenems are often used as a last resort antibiotic to treat severe bacterial infections when other antibiotics fail. The rise in carbapenem resistance compromises our ability to combat life-threatening infections, leading to higher morbidity, mortality, and healthcare

costs. Carbapenemases, enzymes produced by certain bacteria, break down carbapenems, rendering them ineffective. These enzymes can spread between bacteria, exacerbating the problem. The increasing prevalence of carbapenem-resistant bacteria poses a significant threat to public health by limiting treatment options and increasing the risk of uncontrollable outbreaks. As we analyzed the microbiome of such a large number of people, we decided screening their microbial inhabitants for genes encoding carbapenem resistance is an important step in regards to monitoring the spread of such resistance genes.

Deeming the 729 species level bins as "new" also does not sit right. The lack of mapping to known species always happens, most likely due to lack of coverage to explain all scaffolds, incomplete catalogs, etc. To label them "new" would necessarily include comparison to other cohorts of similar design.

Reviewer 2 Comment 5

We acknowledge the urgent shortcoming highlighted by both reviewers regarding the lack of appropriate analysis in the SGB evaluation section. GTDB is a database that provides a selection of representative microbial genomes. This collection is built by collecting a vast number of genomes from data repositories such as RefSeq or GenBank and filtering the genomes using similarity measures. Unfortunately, it appears that GTDB currently does not scrape genomes from resources such as the European Nucleotide Archive. It follows that some representative genomes are missing from this collection that have already been reported. While our analysis showed that GTDB is by far the most complete reference database (**Supplemental Figure 4b**), the reviewer is right to point out that we missed a considerable amount of already documented MAGs. By extending GTDB with three large scale studies of similar design, we reduce the number of falsely reported *unmatched* genomes by 146, and hope to comply with the reviewer's definition of *new*.

We have taken significant steps to rectify this issue and have thoroughly revised the manuscript accordingly. We believe that the changes made not only address the concerns raised by Reviewer 1 Comment 2 but also adequately satisfy the concerns voiced by Reviewer 2 in this comment. We thank the reviewer for voicing these concerns and refer to our response to Reviewer 1 Comment 2 for more details on our updated methodology. Updated numbers on detected SGBs can be found in the abstract. Further modifications to results and methods can be found in lines 311ff and lines 516ff, respectively.

Using biosynthetic gene cluster predictions, the authors identify >28,000 BGC present in the metagenomics datasets, with ~1,000 showing statistically significant signals with disease. The analysis then abruptly stops and any further information about these BGCs is missing: which species encode them, what are the BGCs families and are there any predicted molecular products. Do they provide further implications for community structures? The authors fail to mention a landmark study seeking BGCs in human metagenomics datasets (PMID: 25215495) that provides a blueprint for such investigations. The authors deem that "presented data strongly advocate for prioritizing future investigations", with no guidelines or examples of signals warranting such investments.

Reviewer 2 Comment 6

The reviewer raises several valid points and questions which is further underlined by the fact that Reviewer 1 asked for similar additional information. To better structure our response, we extracted several key points from the reviewer's comment which we respond to, on a point-by-point basis.

Host Species: As we predicted BGCs directly from metagenomic assembled fragments before the definition of SGBs that frequently remove many fragments from further downstream analysis, a reliable assignment of BGCs to derived species proves difficult and partially unreliable (line 338). Nevertheless, we performed taxonomic assignment for the individual 814 fragments defining the 1050 associations. For the taxonomic assignment, we used Kraken2. Seven fragments were not assigned down to species level. We visualized the remaining fragments as a new **Figure 4C** which we here copy for the convenience of the reviewer.

Figure 4C: Predicted host species distribution of the assembled DNA fragments where significantly associated core biosynthetic genes reside.

BGC families: We hope we adequately responded to this concern in the answer to Reviewer 1 Comment 5 (line 339ff).

Predicted molecular products: We hope we adequately responded to this concern in the answer to Reviewer 1 Comment 5 (line 339ff).

Implication on community structure: Indeed, the question for BGCs associated with a disease is, whether the association is not mediated through a change in microbiome composition in response to the expression of a BGC in the community. Unfortunately, with over 28,000 thousand BGCs and 1500 measured species, it is difficult to isolate a clear signal from our dataset which can uniquely link a differential coverage in one individual BGCs to an abundance change in another species. We could cherry-pick a correlative association between changing BGC coverage and species abundance. This is especially trivial considering that the BGC abundance should correlate with host species abundance. However, we do not believe that such an analysis would do justice to the complexity of the research question or follow good scientific practice. Instead, formal analysis including adjustment for correlative effects among BGCs, adjustment of correlative effects among species hosting the BGC, and adjustment for cohort properties should be performed. However, due to the large number of candidate combinations, we currently do not see a statistically feasible approach capable of deciding whether the presence or absence of a BGC has implications on the abundance of individual community members given our current metagenomics sequencing data. In light of all these challenges and concerns, we believe that the investigation of this research question lies well beyond the scope of this manuscript.

PMID 25215495: We added the missing reference in the manuscript which can be found in line 464.

Prioritization: We hope we adequately responded to this concern in the answer to Reviewer 1 Comment 6 (line 355).

We support the overwhelming majority of the points raised by the reviewer. Accordingly, we also added them to the manuscript. Because of the complexity, we avoid copying these comments to the point-by-point response. On this note, we want to take the opportunity to thank the reviewer for the suggested improvements to our work.

In summary, we deem the current version of the manuscript premature for publishing. However, we are encouraged by careful processing and quality controls that promise identification of species and pathways strongly predicting the mentioned conditions.

Reviewer 2 Comment 7

The comments of both reviewers substantiated our work. After 5 years of work we are convinced that the project and data as well as first results are of substantial value for the research community. It is our firm belief that the scientific rigor and depth of analysis presented in this paper align with the standards upheld by Nature Communications but ask for understanding that in-depth functional work, taking from our experience 1-2 years, is beyond the scope of the first manuscript describing the IMAGINE study. Through this publication, we not only make the data accessible to other scientists in the community, but we also provide a comprehensive description with several significant findings at different levels, including species abundance and analysis.

The overall complexity of the dataset makes it prohibitive to fully elaborate on every aspect in every detail. Therefore, we focused on providing a clear, comprehensive, and cohesive analysis of biosynthetic gene clusters, which are of central interest to a large part of the research community. It is important to note that this report serves as a foundation for months, if not years, of further experimental work based on the results presented here.

Lastly, while we largely agree with the reviewer and want to sincerely thank for engaging and contributing to our work, we also appreciate the open and constructive scientific discourse. Their critical feedback reassures us that our data quality and analysis have been rigorously evaluated and are deemed valuable to the scientific community.

References

- 1 Parks, D. H. *et al.* GTDB: an ongoing census of bacterial and archaeal diversity through a phylogenetically consistent, rank normalized and complete genome-based taxonomy. *Nucleic Acids Research* **50**, D785-D794 (2021). <https://doi.org:10.1093/nar/gkab776>
- 2 Pasoli, E. *et al.* Extensive Unexplored Human Microbiome Diversity Revealed by Over 150,000 Genomes from Metagenomes Spanning Age, Geography, and Lifestyle. *Cell* **176**, 649-662 e620 (2019). <https://doi.org:10.1016/j.cell.2019.01.001>
- 3 Almeida, A. *et al.* A unified catalog of 204,938 reference genomes from the human gut microbiome. *Nat Biotechnol* **39**, 105-114 (2021). <https://doi.org:10.1038/s41587-020-0603-3>

- 4 Gounot, J. S. *et al.* Genome-centric analysis of short and long read metagenomes reveals uncharacterized microbiome diversity in Southeast Asians. *Nat Commun* **13**, 6044 (2022). <https://doi.org:10.1038/s41467-022-33782-z>
- 5 Beghini, F. *et al.* Integrating taxonomic, functional, and strain-level profiling of diverse microbial communities with bioBakery 3. *Elife* **10** (2021). <https://doi.org:10.7554/eLife.65088>
- 6 Terlouw, B. R. *et al.* MIBiG 3.0: a community-driven effort to annotate experimentally validated biosynthetic gene clusters. *Nucleic Acids Res* **51**, D603-D610 (2023). <https://doi.org:10.1093/nar/gkac1049>
- 7 Kautsar, S. A., Blin, K., Shaw, S., Weber, T. & Medema, M. H. BiG-FAM: the biosynthetic gene cluster families database. *Nucleic Acids Res* **49**, D490-D497 (2021). <https://doi.org:10.1093/nar/gkaa812>
- 8 Hirsch, P. *et al.* ABC-HuMi: the Atlas of Biosynthetic Gene Clusters in the Human Microbiome. *Nucleic Acids Res* **52**, D579-D585 (2024). <https://doi.org:10.1093/nar/gkad1086>
- 9 Blin, K., Shaw, S., Medema, M. H. & Weber, T. The antiSMASH database version 4: additional genomes and BGCs, new sequence-based searches and more. *Nucleic Acids Res* **52**, D586-D589 (2024). <https://doi.org:10.1093/nar/gkad984>

Reviewers' Comments:

Reviewer #1:

Remarks to the Author:

Thank you for sharing your carefully written rebuttal. I am very much satisfied with the changes made, and I believe the manuscript now looks in a very good shape. I look forward to seeing it published.

RESPONSE TO REVIEWERS' COMMENTS

Reviewer #1 (Remarks to the Author):

Thank you for sharing your carefully written rebuttal. I am very much satisfied with the changes made, and I believe the manuscript now looks in a very good shape. I look forward to seeing it published.

We thank the reviewer for their valuable feedback.